# Evaluating Urban Vitality of Street Blocks Based on Multi-Source Geographic Big Data: A Case Study of Shenzhen

**DOI:** 10.3390/ijerph20053821

**Published:** 2023-02-21

**Authors:** Ziyu Wang, Nan Xia, Xin Zhao, Xing Gao, Sudan Zhuang, Manchun Li

**Affiliations:** 1Jiangsu Provincial Key Laboratory of Geographic Information Technology, School of Geography and Ocean Science, Nanjing University, Nanjing 210023, China; 2Department of Geographic Information Science, Nanjing University, Nanjing 210023, China; 3Collaborative Innovation Center for the South Sea Studies, Nanjing University, Nanjing 210023, China; 4Collaborative Innovation Center of Novel Software Technology and Industrialization, Nanjing University, Nanjing 210023, China

**Keywords:** urban vitality evaluations, machine learning, random forest, open-source data

## Abstract

Urban vitality is the comprehensive form of regional development quality, sustainability, and attractiveness. Urban vitality of various regions within the cities has difference, and the quantitative evaluation of urban vitality within the cities can help guide to future city constructions. Evaluation of urban vitality needs the combination of multi-source data. Existing studies have developed index method and estimation models mainly based on geographic big data to evaluate urban vitality. This study aims to combine remote sensing data with geographic big data to evaluate urban vitality of Shenzhen at street block scale and build the estimation model by random forest method. Indexes and random forest model were built, and some further analyses were conducted. The results were: (1) urban vitality in Shenzhen was high in the coastal areas, business areas, and new towns; (2) compared to indexes, the estimation model had advantages of more accurate results, combination of various data, and the ability to analyze feature contributions; and (3) taxi trajectory, nighttime light, and housing rental data had the strongest influence on urban vitality.

## 1. Introduction

The term “urban vitality” was widely accepted to describe the quality, sustainability, attractiveness of spaces that attract people for different types of activities to produce diverse city life [1,2]. Higher urban vitality usually means stronger attraction to people, the ability to support demands of different people, and the potentiality of development [3,4]. Improving the urban vitality is significant to the urban development and human lives, and the evaluation of current situation is the basic work [2,5]. Meanwhile, different areas within the cities have different functions and orientations, leading to different developing patterns and velocities, and urban vitality [6]. Studying on urban vitality of different areas within cities has received considerable attention in China [7]. The quantitative evaluations of intracity urban vitality with different study units, such as street blocks, communities, districts, and grids, can help to understand the regional and overall development conditions, and can guide the future city activities, planning, and constructions [5,8].

As a comprehensive concept, the quantitative evaluation of urban vitality should consider various aspects [9,10], mainly including density, diversity, accessibility, interaction, and permeability [11,12,13,14]. These varying factors cannot be summarized by a small amount of data; therefore, combining multi-source data is necessary [15]. The most direct representation of urban vitality is based on the population distribution data or human activity data, such as population census data and mobile phone signaling data [16]. However, these data are usually hard to obtain owing to various limitations, such as data privacy and collection frequency. Therefore, other substitutive open-access big data with geolocation information are developed to evaluate urban vitality, including point of interest (POI) data [3,5], social media check-in data [17,18], taxi GPS data [19], takeaways data [20], location-based service data [21,22], street view images [23], heat map data [24], and building foot-print data [10]. These different kinds of geographic big data can represent urban vitality from various perspectives of human activity footprint. For example, POI data emphasizes on diversity and density of facilities and activities [3], social media check-in data stresses people’s activities and density of people [25], and taxi GPS data reflects the flows of people and accessibility of some areas [19].

To combine the data with abundant information and different perspectives, two methods were developed: indexes and estimation models. The former uses some processing results of the data (e.g., the original value, the average value, and the ratio of the value to the area of the research unit) to build urban vitality indexes. Usually, several representative indexes are built based on a small amount of data; sometimes researchers build index systems [6,26,27]. Building indexes is simple and adaptable to different conditions of data and research topics [23]; however, advantages of the diversity and high dimensions of geographic big data have not been realized, and the accuracy of the results is not satisfied [28]. Moreover, utilization of remote sensing data is not sufficient, which can reflect information of land cover/use, construction situation, etc. The nighttime light remote sensing data were also paid more attention for urban vitality evaluations, which can reflect the intensity of artificial nighttime light [29]. However, remote sensing data cannot directly represent the intensity of human activities, preferably combinations with other geographic big data [30].

Estimation models mostly depend on the relationship between reference data and input data to build complex model. Researchers have used the entropy model [31], regression model [32,33], coupling model [34], Technique for Order Preference by Similarity to Ideal Solution (TOPSIS) model [35], panel fixed effect model [36], and spatially explicit indices model [12] in urban vitality studies. Geographic big data such as POI data, social media check-in data, taxi trajectory data, etc., can be combined with remote sensing data in these estimation models, superior to the simple-structured index method [37]. However, because of the differences in semantics, coverages, data quality, data values, and organizing forms between remote sensing data and geographic big data, suitable estimation models to combine these two kinds of these data are still difficult for urban vitality evaluations [36]. Moreover, existing researches only consider some commonly used data types, and potentials of other data are neglected. For example, vegetation index data could reflect the vegetation coverage and growth conditions, and housing rental data could reflect the residence demand and economic conditions, which both had strong correlations with human activities and were seldom considered [38]. Meanwhile, the absence of these data could also be attributed to the weak simulation ability of the above-mentioned models.

Machine learning is a powerful tool for finding patterns in data, analyzing the characteristics of data and objects, and combing large varieties of data [39,40], especially useful for multi-dimensional and multi-semantic data, such as geographic big data [41]. Considering these advantages, machine learning is now widely used in geographic studies [42,43], and is also applied to promote the performance of urban vitality evaluation by remote sensing data and geography big data [44,45]. By combing s a certain number of weak learners, the ensemble learning model was further proposed to obtain more reliable and accurate results [46]. The Random Forest model (RF) is a kind of widely used ensemble model based on decision tree algorithm and had the advantages of simple operation, fast calculating speed, credible results, and the ability for providing many parameters for subsequent analysis [47,48,49]. The selection of reference data and spatial quantization of input features are essential for the construction of RF model [41]. Some of the most intuitive data, like mobile phone signaling data and POI data, can be served as the reference data for the strong correlations with urban vitality and data reliability [50]. With the help of open data platform, web crawler, and network API, etc., open-access geographic big data can be easily acquired. Different from the standard raster data format of remote sensing data, the spatial quantization of geographic big data within cities still needs specific analysis for data types and research basic unit [4].

Over the past few decades, China’s urbanization has increased rapidly; the urban area has increased from <7500 km^2^ in 1980 to >60,000 km^2^ in 2020 [8]. Cities are expanding rapidly with several problems in some urban areas, and how to combine multi-source data and machine learning method to quantitatively evaluate the urban vitality is still a problem [51]. As a fast-growing megacity, Shenzhen was selected as the study area for evaluation of urban vitality within the cites, and the aim of this was to (1) build indexes to evaluate urban vitality, (2) build a machine learning model to combine multi-source data for urban vitality evaluation, and (3) discuss about different methods and try to find a promotable way to evaluate urban vitality.

## 2. Study Area and Data

### 2.1. Study Area

Shenzhen in Guangdong province was selected as the study area (Figure 1). Shenzhen is located at Pearl River Delta, on the south coast of China, north of Hongkong (113°46′–114°37′ E, 22°27′–22°52′ N). Shenzhen is one of the earliest special economic zones of China. As one of the most prosperous megacities in China, Shenzhen spans an area of 1997.47 km^2^. It has 10 districts (Luohu, Futian, Yantian, Nanshan, Baoan, Longgang, Longhua, Pingshan, Guangming, and Dapeng); 74 streets; and 787 communities. Shenzhen’s GDP reaches 2800 billion China yuan, ranking third among all cities in China. Shenzhen has 17.63 million permanent residents, with an increase of 530,000 over the previous year, and the population’s natural growth rate is 1.58%. With 9.5 million migrant workers, over half of permanent residents, Shenzhen has a total of about 27 million people. From 2011 to 2019, Shenzhen hosted approximately 12 million yearly tourists. The China Urban Vitality Research Report of 2020 and 2021, composed by Baidu, considered Shenzhen as a city with the highest urban vitality in China. Overall, Shenzhen is one of the most energetic cities in China, with a high development speed.

### 2.2. Data and Data Preprocessing

The data for this study include four types: OpenStreetMap data, remote sensing data, geographic big data, and urban vitality reference data. The data were downloaded from some websites (See Table 1). All data were processed under WGS84 geographic coordinates system.

#### 2.2.1. OpenStreetMap Data

OpenStreetMap (OSM) is an open-source map website, providing detailed information of roads and facilities around the world. The data was processed using the OpenStreetMap toolbox in ArcGIS10.7. The data has the property of road categories with WGS84 coordinates, such as primary roads, secondary roads, roads in residential areas, etc. The primary and secondary roads were selected, and 6694 blocks were divided accordingly [52]. All these blocks covered built-up areas of 800.15 km^2^ in Shenzhen, but not forest areas (Figure 2). The largest block is Baoan International Airport, whose size is approximately 18.46 km^2^, which is obviously larger than other blocks. The second largest size of the blocks is 4.10 km^2^, and the minimum size of the blocks is 7439 m^2^. The average size of the blocks is 119,739 m^2^.

#### 2.2.2. Remote Sensing Data

Remote sensing data included land use data, nighttime light data, and moderate-resolution imaging spectroradiometer (MODIS) normalized difference vegetation index/enhanced vegetation index (NDVI/EVI) data.

Land use data came from Globalland30, which was a global land cover data developed by Ministry of Natural Resources of the People’s Republic of China, with a spatial resolution of 30 m. Compared with other global land cover data, such as Global Land Survey data, CCI Land Cover V2 data, Globalland30 has the advantages of high resolution and easy accessibility. The data included the 2010 and 2020 versions, and data of 2020 version were obtained. Globalland30 includes 10 land use types: cultivated land, forest, grassland, shrubland, wetland, waterbody, tundra, bare land, artificial land, and glacier. The cultivated land, grassland and forest were merged as ecological land, and Shenzhen only had land use types of ecological land, artificial land and waterbody.

Nighttime light data was obtained by satellite sensors recording received artificial and natural light intensity during the nighttime, and larger pixel values indicated higher light intensity and human activities. Nighttime light images were from Luojia1 satellite by Wuhan University. Data of November 2018 was obtained with a spatial resolution of 130 m.

Moderate-resolution imaging spectroradiometer (MODIS) is a remote sensing instrument from U.S. Normalized difference vegetation index (NDVI) and enhanced vegetation index (EVI) are two vegetation index products from MODIS, reflecting vegetation cover conditions on the surface. The higher NDVI/EVI values represented the denser and lusher vegetation cover [53]. This study obtained data of December 2018 with a spatial resolution of 250 m.

#### 2.2.3. Geographic Big Data

This study used three types of geographic big data: taxi trajectory data, Weibo check-in data, and housing rental data.

The study gathered the taxi trajectory data in November 2019. Original data included taxi points obtained by satellite positioning and was stored in a txt file. The attributes included the identify ID, the operation times, the coordinates, the velocities, and whether the taxi was carrying passengers in Shenzhen. The Pandas library in Python was used to retrieve and remove abnormal data, find the start point and the end point of each journey, connect the start and end points to create taxi trajectories, and calculate Euclidean distance as the length of each trajectory [54]. After preprocessing, the study obtained 7,874,790 taxi trajectories.

Weibo check-in data was obtained from Weibo using a Python web crawler, Weibo is a popular social media platform in China with millions of users. The crawler, developed by the weiboSpider module, gathered 1,212,051 check-in data in 2019 with abundant attributes, including time, location, and users’ information.

Housing rental data of 2019 was obtained from Anjuke, a commonly used online real estate information release platform in China. The data was retrieved using a Python web crawler with requests and urllib modules. The crawler gathered information on 7714 real estate properties in Shenzhen and included attributes like ID, name, location, rental amount, and rental type. Rental types include residential rents, commercial rents, and other rents. The rental amount is expressed as the rental price per square meter.

#### 2.2.4. Urban Vitality Reference Data

Point of interest (POI) data is a commonly used index for evaluating urban vitality and is taken as reference data for urban vitality. The POIs are mainly the public facilities and business spots in the city; therefore, the number and the density of POIs reflect the level of economic activities somewhere, and then reflect urban vitality of different areas within the cities [24]. The data was obtained from Baidu Map (https://map.baidu.com), a popular Internet map in China. The study gathered data from 2020, including about 13 million POIs in Shenzhen.

## 3. Methods

This study comprised three main steps: (1) data processing and features extraction, (2) construction of urban vitality index, and (3) construction of urban vitality estimation model.

### 3.1. Data Processing and Features Extraction

The 6694 blocks in Section 2.2 were taken as research objects and data was calculated on every block. Redundant data were removed according to fields and locations.

Reference (POI) data, Weibo check-in data, and housing rental data represent point data. All data points were segmented into the corresponding blocks. Then, the POI data was processed by counting the total number of POIs in each block, and Weibo check-in data was divided into the daytime and nighttime parts and counted separately. The average values and summary values of all types of rents, residential rents, and commercial rents were calculated. The number of POI data and Weibo Check-in data were then divided by the area of the blocks to reduce the effect of different block areas.

The OSM road data and taxi trajectory data represent line data. When processing OSM road data, the length of all roads (i.e., primary and secondary roads) was calculated. Next, the density of the roads was calculated. All trajectories whose starting points in the block were found and had their sum length and average length calculated by ArcGIS10.7. The same was done for the trajectories whose ending points were in the block.

Nighttime light data and MODIS NDVI/EVI data represent raster data. The first step was to judge which block each pixel of raster data belonged to. Then, if a pixel’s geometric center was located in a block, it would be assigned to the block. The average values and summary values of all pixels located in the blocks were then calculated by ArcGIS10.7.

When processing land use data with raster format, four results were calculated for each block: area proportion of urban land pixels; area proportion of water body pixels; area proportion of ecological land pixels (including cultivated land, forest land, and grassland); and Shannon diversity index (SHDI).

To reflect the density, diversity, accessibility, interaction and permeability aspects of the urban vitality, a total of twenty-seven features were constructed based on the abovementioned four types of data. Table 1 shows those features and the features’ data source, and explains the building methods of the features. According to the data sources of the features, these 27 features can be divided into seven categories: taxi trajectory features, Weibo check-in features, housing rental features, land use features, OSM road features, nighttime light features, MODIS NDVI/EVI features.

### 3.2. Construction of Urban Vitality Index

Indexes were built based on the features listed in Table 1. Building indexes need expressions to be determined manually, therefore, indexes can only deal with low dimensional features. Therefore, no more than one feature from each data source could be selected. This study tried to select representative features from all 27 features in Table 1, where only one feature from the same data source could be retained. The Pearson correlation coefficients (PCCs) between reference data and all features were calculated.

Features *HR_price_all* and *Luojia_mean* had the highest PCCs with the reference data, while all OSM features, MODIS features, and land use features had the lowest PCCs. Taxi trajectory features and Weibo check-in features had mid-level PCCs. Therefore, features *HR_price_all* and *Luojia_mean* were selected, then features *Taxi_total_mean* and *Weibo_sum*, separately from the taxi trajectory and Weibo check-in data, were added. These selected features were normalized to [0, 1], then indexes were built.

The PCCs of these four features with the reference data were positive, therefore, indexes were built by adding them up and multiplying them together. The weighting method had not been applied for indexes because the weights were difficult to determine. Moreover, the weighting would cause some unexpected errors [55,56]. In total, two indexes were built: C+W+L+T (Index 1) and C×W×L×T (Index 2), where *C* represented *HR_price_all*, *W* represented *Weibo_sum*, *L* represented *Luojia_mean*, *T* represented *Taxi_total_mean*.

### 3.3. Urban Vitality Estimation Model

Random Forest is an ensemble learning model based on decision trees. The data will be divided into training parts and test parts, to obtain weak learners from the training data to generate a strong result. Finally, the result will be examined by test data.

The data were divided by the train_test_split, a module in Skleran library in Python, with a proportion of 30% and 70% separately. The RF model was run by the RF module in Sklearn. Before setting the RF model, PCCs of every feature in Section 3.1 with other features were calculated to find and exclude redundant features. Next, an RF learning model was built, which took POI as dependent variable and kept the remained features as independent variable. Next, the RF calculated its results.

Setting parameters is essential for an RF model. The main parameters of RF included N_estimators, the maximum number of the weak learners obtained from the original dataset; Max_features, the number of features selected in each decision tree; Max_depth, the maximum depth of a decision tree; Min_samples_leaf, minimum number of samples for leaf nodes; Min_samples_split, minimum number of samples for node splitting. N_estimators and Max_depth are the most important parameters, for they decide the magnitude of overfitting and complexity of calculation. When being set, every parameter is changed according to a footstep. Table 2 shows the setting ranges and the adjusting footsteps of parameters in RF according to existing studies [57]. The optimal values of the parameters were calculated by GridsearchCV module from Sklearn library in Python [54].

## 4. Results

The spatial distribution pattern of the POI reference data is shown in Figure 3, with natural breaks (Jenks) classification of the five urban vitality levels, including Very High, High, Medium, Low, and Very Low. The areas with the highest urban vitality were located in Nanshan, Luohu, Futian, Baoan, middle of Longhua, and the middle of Longgang. The overall distribution pattern of urban vitality was high in the western and southern regions and low in the eastern and in the northern regions. There were 533 blocks at the Very High, and High level of urban vitality, about 7.96% of total 6694 blocks and 6.91% of the total area. On contrast, 5153 blocks were at the Low and Very Low level, about 76.98% of the total 6694 blocks and 78.48% of the total area.

### 4.1. Evaluation Result of Urban Vitality Index

Indexes in Section 3.2 were compared with Figure 3 by calculating the PCCs with the reference data and visual comparison. The results of the indexes were shown in Figure 4. The evaluation values of Index 1 were overall higher than Index 2. Indexes 1 and 2 correctly reflected the areas with high urban vitality in Nanshan, Futian, Luohu, and middle of Longhua. Moreover, the performance of Index 1 and Index 2 in Longgang, the middle of Baoan, and Dapeng were close. However, Index 1 wasn’t able to find the areas with low urban vitality in Pingshan, and Index 2 failed to find the areas with high urban vitality in north of Baoan. Both of the indexes obtained higher values in Yantian, Guangming, north of Longhua, and Baoan International Airport. The distribution of reference data showed a cluster of high values and a gradual decrease in these high-value areas out. Conversely, urban vitality by the index method did not show such a cluster but rather fragments. It could be inferred that the index method was able to roughly evaluate urban vitality and find high and low-value areas; however, some mistakes might arise in certain areas, and an index might be overall higher or lower.

### 4.2. Evaluation Result of Urban Vitality Estimation Model

The PCCs between different pairs of features and between features and reference data were calculated (Figure 5). Among five taxi trajectory features, the PCCs among three average features and between two summary features were high, but considering that in-features and out-features might have different influence, only *Taxi_total_mean* was excluded as a redundant feature. Similarly, *Weibo_sum* was excluded because the other two Weibo check-in features were retained to explore the difference between daytime and nighttime check-in data. The PCCs among three housing rental features were high, but features *HR_price_com* and *HR_price_res* had obviously lower PCCs with reference data than feature *HR_price_all*, Therefore, those two features were both excluded. The PCCs of primary road features and secondary road features from OSM data were high, therefore, the secondary road features were excluded. The two NDVI features were excluded for similar reason with EVI features. In summary, eight redundant features were excluded, including: *Weibo_sum*, *HR_price_res*, *HR_price_com*, *Taxi_total_mean*, *OSM_sec_len*, *OSM_sec_den*, *NDVI_mean*, and *NDVI_sum*, and a total of 19 features were retained.

By GridsearchCV module in Python, the optimal values of the parameters in the RF model were obtained. The optimal value of N_estimators was set as 135, 7 for *Max_features*, 16 for Max_depth, 3 for Min_samples_split, and 2 for Min_samples_leaf. The RF model was run on these parameters, with the result shown in Figure 6. Compared with the reference data in Figure 3, the RF model was able to correctly reflect areas with high and low values. It correctly found high-value areas and low-value areas. Moreover, in Yantian, Guangming, north of Longhua, and Baoan International Airport, the evaluation results of RF model were closer to the reference data than the indexes. The RF model showed the cluster of high values and the gradual decrease in these high-value areas out as the reference data did. The number of the blocks at Very High and High level of urban vitality from was 577, about 8.62% of total blocks, 8.80% of total areas. For the Low and Very Low level, 4956 blocks contributed to about 39.88%, and 33.35% of total areas, respectively. These statistics result by the RF model was close to the reference data.

Altogether, urban vitality estimation model performed better than the indexes. High values were mainly concentrated in Nanshan, Luohu, and Futian. Moreover, some areas in Longhua and Longgang were also correctly reflected.

Based on the evaluation results of the urban vitality index and the estimation model, it could be seen that Nanshan, Luohu, and Futian were areas with the highest urban vitality in Shenzhen. Baoan, Longhua, and Longgang were secondary. Dapeng and Pingshan had the lowest urban vitality, while the other areas were mid-range. Based on the functional area division of Shenzhen, it was speculated that urban vitality reached a high level in coastal prosperous areas, business zones, newly developed city areas, and areas full of high-tech companies, while ecological areas, forest parks, and some inland areas had low urban vitality.

## 5. Discussion

### 5.1. Comparison of Urban Vitality Index and Urban Vitality Estimation Model

#### 5.1.1. Estimation Accuracy of and Method Limitation Analysis

The accuracy of the indexes and estimation model has been compared. Figure 7 shows the scatter plots of normalized (0–1) Index 1, Index 2, and the RF model with the reference data. The PCCs of Index 1, Index 2, and the RF model with reference data were calculated and found to be 0.5727, 0.5406, and 0.7837, respectively. Here, the RF model had a strong positive correlation. Moreover, Figure 7 shows the confidence intervals (CI) of 5% of the evaluation results. The CI of the RF model is smaller than that of indexes 1 and 2, indicating a better evaluation accuracy for estimation model. Therefore, the estimation model’s evaluation accuracy is better than the indexes. Both the indexes and the estimation model can correctly find high-value areas. However, the estimation model was more closed to the reference data in low-values areas (Figure 4 and Figure 6). It can be judged that the estimation model performed better than the indexes.

Four reasons can explain why the estimation model performed better, which is also the limitations of index method: (1) the estimation model used more features. The estimation model used 19 features, while the indexes only used 4 features; (2) the estimation model combined big data and remote sensing data. Both of big data and remote sensing data can evaluate urban vitality, however, the indexes were unable to combine them effectively. While the estimation model can combine the merits of big data and remote sensing data to obtain better results; (3) the estimation model can perform weighting of the features. Indexes combined the data with no difference, however, some features’ influence on urban vitality were strong while other features’ influence were weak. The indexes were difficult to find such differences. On the contrary, the estimation model can obtain the weights for the features to control their contributions to the result, making the result better; (4) the estimation model could select and combine features more objectively. The number of features used in the indexes was limited, only four features were selected by PCCs with the reference data. However, several excluded features also had high PCCs (Figure 5), which might have good effect on evaluation. In contrast, the estimation model could contain features as many as possible, just after excluding obviously abundant features. In addition, there were some subjective factors in building the indexes by trial-and-error which may influence the evaluation accuracy. While the estimation model was built by decision trees from the training data, and the training data was obtained by the train_test_split module in the Sklearn library, with no human intervention in the whole process.

However, the estimation model also has limitations. The main limitation is the in-fluence of the data quality, which was mainly the different geographic big data with different data source, data structures, and processing methods. Meanwhile, remote sensing data and OSM data are accessible, while geographic big data may not be easily acquired in other study area, such as taxi data. The combination with more kinds of geographic big data will reduce the transferability of out method. Moreover, different study areas have various urban layout, land use planning, and development design, leading to the difference of input features in data quality, performance and spatial pattern. The selection of the representative features by feature contribution (Section 5.2 for details) may greatly depend on the characteristics of the study area, increasing the uncertainty of our method.

#### 5.1.2. Case Analysis on the Difference of Two Methods

To explain the four reasons above, some sample plots were selected to make the case analyses. The sample plots should locate in the areas where the results of the indexes and the estimation model differentiate. Therefore, an overall comparison of the indexes and the estimation model was conducted, which showed that in commercial areas, residential areas near commercial areas, and central areas of cities, the evaluation values of the two methods were very close. For example, in Luohu and Futian, indexes 1 and 2 and the RF model had no obvious differences. However, in other areas, the values of the estimation model were closer to the reference data. The three most typical cases were: industrial and transportation areas, residential areas on urban fringe, and urban villages.

(1) Industrial and transportation areas. Yantian Port, the largest port in Shenzhen, is located in the south of Yantian (Figure 8). The satellite image is full of containers with few POIs and business areas. This area is mainly for cargo distribution and throughput, with few other kinds of activities and small population. Thus, the urban vitality of Yantian Port should be low. However, the port works 24/7, and the nighttime lights are so intense that the feature *Luojia_mean* here is conspicuously high. As a result, both indexes 1 and 2 obtained high values, while RF’s evaluation was normal. In Baoan International Airport, a similar phenomenon occurred because the feature *Weibo_sum* was high for many people preferring to check in at the airport [58].

(2) Residential areas on urban fringe. Baoan district has a population of 4.47 million, ranking first in all eleven districts in Shenzhen. Baoan is also an important economic and industrial zone, with concentrated residential areas, dense road networks, and larger population inflow. The reference data in the north of Baoan shows the mediate or high urban vitality (Figure 8). However, as the residential area on the edge of the city, business and economical facilities are few in these regions. Therefore, plenty of activity-related geographic big data are missing, such as Weibo check-in data, leading to the low level of relevant values. Conversely, remote sensing data can show normal values to represent human activities in these areas, such as nighttime light data. Thus, with the combination of these remote sensing data, the estimation model could perform better.

(3) Urban Villages. Baishizhou Street, one of the largest urban villages in Shenzhen, locates in the south of Nanshan (Figure 8). Although the population is large, most of the residents are migrant workers. The street layout is very messy, and the facilities are relatively old. Although the urban vitality in Baishizhou Street is low, it locates in the economically developed Nanshan district, with plenty of different city facilities around. The values of the geographic big data are at a high level in Baishizhou Street, while values of some remote sensing data are low, such as nighttime light data. Therefore, the index method obtained higher values than the actual condition in Baishizhou Street, without consideration of different feature weights for remote sensing data and geographic big data, while the RF model obtained a low result, which is more realistic.

### 5.2. Contribution of the Features to Urban Vitality

The RF model provided numerous other parameters in its output, including contributions, or the importance of all features, which can help find the strongest influencing factors of urban vitality evaluations. Figure 9 shows all 19 features’ contribution. For comparison, 19 new RF models were built to compare with the original model. Each new RF model was constructed by removing one of the original 19 input features, and the PCCs of evaluation results with reference data were also calculated for each model (Figure 9).

Among all features, *Taxi_in_mean* and *Luojia_mean* had the highest contributions, which could reflect the accessibility and density aspect of urban vitality respectively. The former’s contribution was over 30%, and the latter also had a contribution of 18.11%. If removed, the PCCs with the reference data decreased obviously (0.6219 and 0.6676, compared to 0.7837 of the original model). That meant taxi trajectory data and nighttime light data had the strongest influence on urban vitality, similar to conclusions of some existing studies [19,29].

*LU_Artificial*, *LU_Ecological*, and *LU_Water*, three land use features, had the lowest contributions (<0.80%). When these features were removed, the PCCs with the reference data showed almost no change (>0.77). The blocks divided by OSM data mainly covered the main built-up areas in Shenzhen. Over 90% of the blocks were mainly artificial lands, and the number of blocks with other land use types is insufficient. The area of Shenzhen is small, as a result, the degree of land development and utilization is relatively high, and the proportion of artificial surface is relatively large. Therefore, influence of these ecological land and waterbody to urban vitality was not obvious in Shenzhen. However, *LU_Shannon* had a contribution of 2.58%, indicating that mixed land use still influenced urban vitality [5,59].

*OSM_pri_len* and *OSM_pri_den* all had low contributions (0.86% and 1.41%). If removed, the PCCs with the reference data were >0.76. Most of primary roads in OSM data are the trunk roads connecting various urban districts and important facilities (e.g., party and government organizations and big industrial areas). These primary roads’ function is connecting districts in the city, or linking different cities. Therefore, primary roads’ function is on city scale, the influence of the primary roads on block scale is weak. However, when considering all road categories, the contribution would be higher. Therefore, a densely distributed road network still influences urban vitality positively [60].

Features from taxi trajectory data performed differently. *Taxi_in_mean* had a contribution of 30.30% while *Taxi_out_mean* had a contribution of 3.25%. Both were average values, yet their contributions largely differed. Therefore, it can be inferred that an area with higher urban vitality might have more driving-in taxi travels. Existing studies showed that when going to more prosperous streets, people prefer taking a taxi at a relatively near distance. Areas with high urban vitality have dense populations, various commercial activities, big shopping centers, and cultural facilities. Therefore, they are more likely to be selected as destinations. The emergence of mobile taxi applications in China made people choose to travel by taxi more often [61]. Two summary features of taxi data had lower contributions than average features but were still high enough (>5%). Therefore, areas with high urban vitality may have more taxi travels.

*Weibo_night*’s contribution was higher than *Weibo_day*. Existing studies have found that Weibo check-in has morning peaks and evening peaks, with the latter being much larger [62]. What’s more, *Luojia_mean* also had a pretty high contribution. That is, nighttime vitality is more important to urban vitality than daytime [5,63,64].

*HR_price_all*’s contribution was 9.34%. Therefore, areas with high urban vitality may have expensive housing rents. Taxi trajectory, nighttime light and MODIS NDVI/EVI features had the same phenomenon: contributions of the average value features were higher than the summary value features. This needs further investigation.

### 5.3. Spatial Pattern of Shenzhen’s Urban Vitality

Urban vitality is a spatial phenomenon, so it may have spatial autocorrelation. The Global Moran Index was selected to analyze the spatial autocorrelation of Shenzhen’s urban vitality [65]. Table 3 shows Global Moran indexes and Z scores of Index 1, Index 2, and the RF model. The spatial weight matrix of the Global Moran Index was based on inverse distance, and the distance was the Euclidean distance.

All indexes and RF’s Z scores were >1.00, showing high confidence of the results. The values were all positive, meaning that urban vitality was of a concentrated distribution, with high and low-value areas clustering together. The Global Moran Index of the RF model was higher than that of Index 1 and Index 2; the evaluation result of the RF model had a stronger autocorrelation.

The Global Moran Index reflects the overall situation of the study area, and only one value is obtained to reflect the overall results, which cannot be visualized. Cluster analysis can yield a more detailed analysis. A cluster analysis, run by Anselin Local Moran’s I model, can find the high-value distribution (HH) and the lower-value distribution (LL) with significant statistical characteristics. Moreover, the low-value aberrant in the high-value area (LH), the high-value aberrant (HL) in the low-value, and areas with no obvious distribution patterns are found [66].

Cluster analysis was exerted on the reference data and RF model results (Figure 10). In the HH areas of Nanshan, Luohu, and Futian, reference data and RF performed similarly. In Longhua, reference data and RF performed similarly as well. In the LL areas, reference data and RF were similar. The main differences were in Baoan and Longgang. In these areas, urban vitality by reference data had large areas of HH, while urban vitality by RF had no significant characteristics. On the whole, urban vitality by RF had more HL and LH aberrant than urban vitality by reference data. It can be judged that high-value areas gather in Luohu, Futian, Nanshan, and Longhua, and low-value areas gather in Dapeng, Pingshan, Guangming, and Yantian. However, in some areas of Baoan and Longgang, further studies are needed.

## 6. Conclusions

A quantitative evaluation of urban vitality is essential to urban management. This study has integrated multi-source geographic data and remote sensing data, and built evaluation indexes and random forest estimation model to evaluate Shenzhen’s urban vitality at a street block scale. Analysis and comparison between two methods were also conducted, and the conclusions could be summarized as follows: (1) both the urban vitality index and urban vitality estimation model could correctly find areas with the highest urban vitality, however, in other areas, the vitality estimation model performed better. The estimation model also had the advantages of high dimension features, more objective processes, and the ability to analyze features’ contributions. (2) The total distribution pattern of urban vitality in Shenzhen was high in the west and low in the east, and high in the south and low in the north. It is influenced by the economic levels and distribution of urban functional zones. High urban vitality areas mainly distributed in coastal areas, business areas, and new towns, such as Nanshan, Futian, and Luohu, also with high aggregated pattern. Moreover, some areas in Baoan, Longhua, and Longgang also had high urban vitality, and low urban vitality areas were mainly in Pingshan and Dapeng. (3) among all features, driving-in travels of taxi trajectory data, nighttime light, and housing rental have obvious positive effects on urban vitality, while road networks, mixed land use, and vegetations and have no weak effects.

Previous studies of urban vitality in Shenzhen mainly used kernel density analysis or some indexes mainly based on geographic big data, such as Baidu Heat Map data, building footprint data, POI data, and mobile phone signaling data [10,24,29]. Similar to the previous studies, this study obtained the highly consistent distribution patterns of urban vitality and large contribution of the geographic big data. Differently, machine learning model, combination with remote sensing data, and quantitative description of feature contribution improved the reliability of the results. In the future study, more types and higher quality of geographic big data could be combined to improve the model performance, such as mobile phone signaling data. Moreover, future studies should also explain the contribution difference of the various features from the same data, such as the average value feature and the summary value feature.

## Figures and Tables

**Figure 1 ijerph-20-03821-f001:**
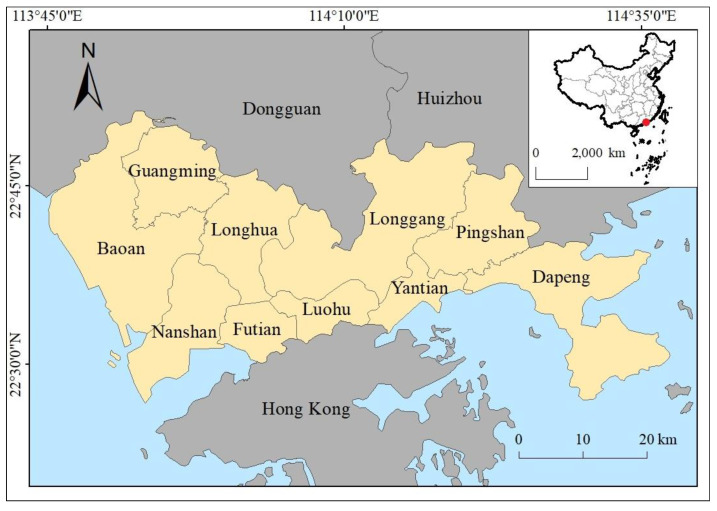
Study area of Shenzhen.

**Figure 2 ijerph-20-03821-f002:**
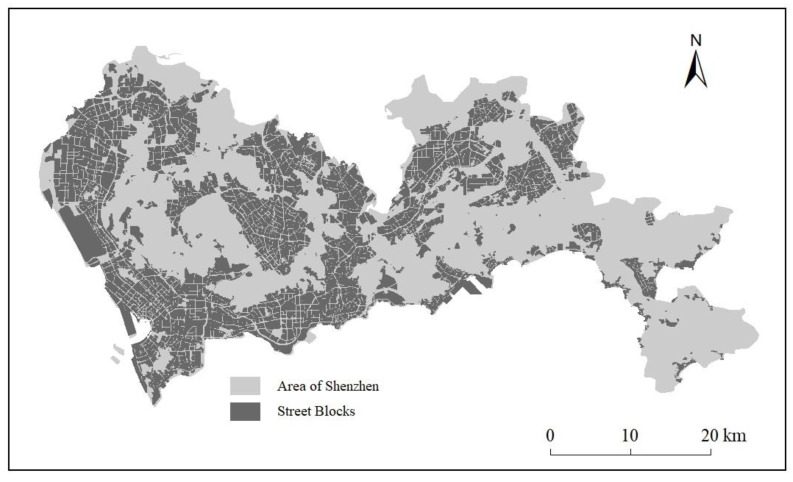
Blocks division result according to OpenStreetMap roads in Shenzhen.

**Figure 3 ijerph-20-03821-f003:**
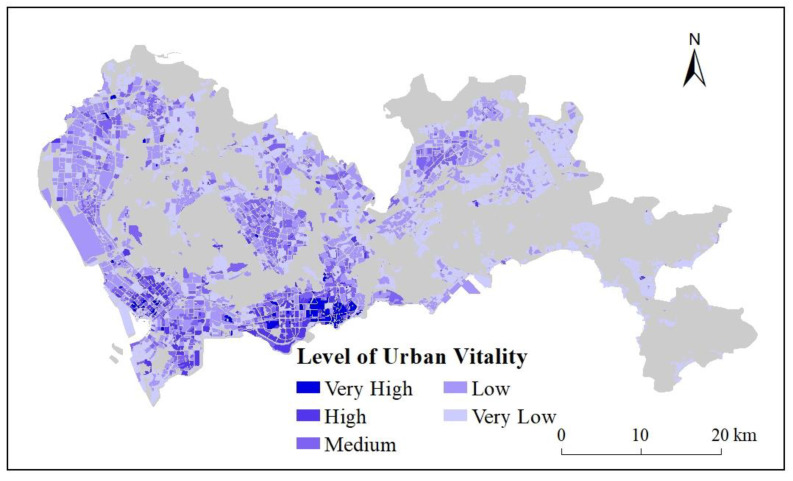
Urban vitality reference data (POI) of Shenzhen.

**Figure 4 ijerph-20-03821-f004:**
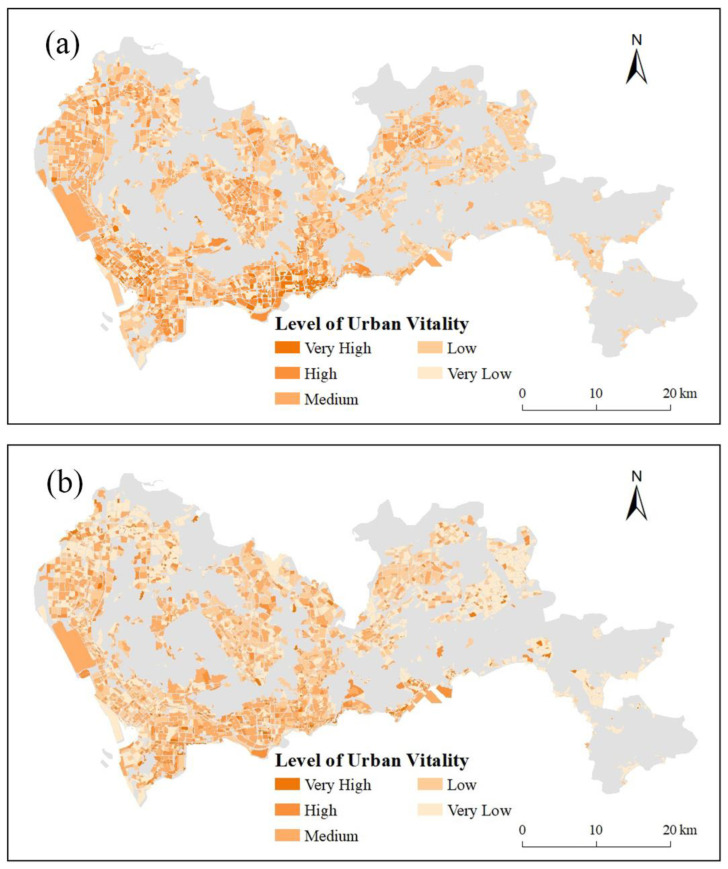
Urban vitality Index 1 ((**a**) C+W+L+T) and Index 2 ((**b**),C×W×L×T). *C*: *HR_price_all*, *T*: *Taxi_total_mean*, *W*: *Weibo_sum*, *L*: *Luojia_mean*.

**Figure 5 ijerph-20-03821-f005:**
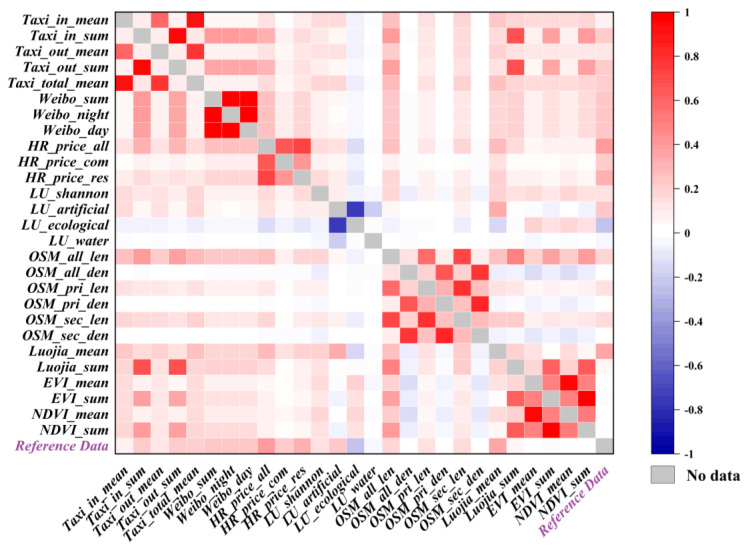
Pearson correlation coefficients of reference data and features.

**Figure 6 ijerph-20-03821-f006:**
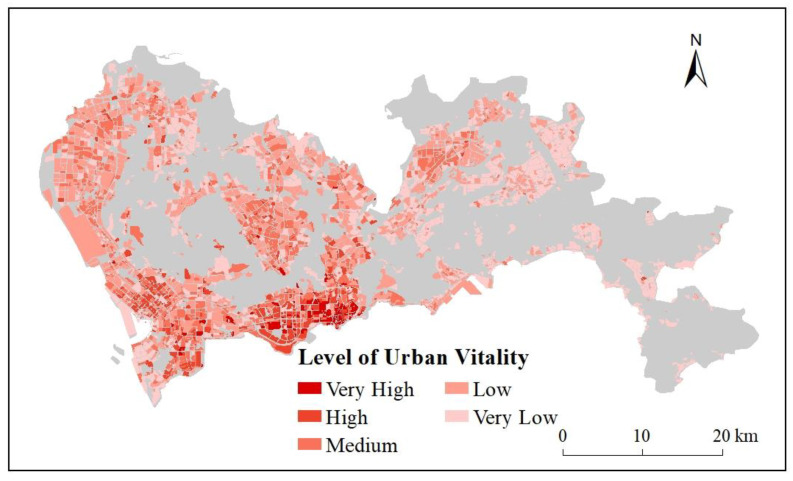
Urban vitality estimation model by Random Forest.

**Figure 7 ijerph-20-03821-f007:**
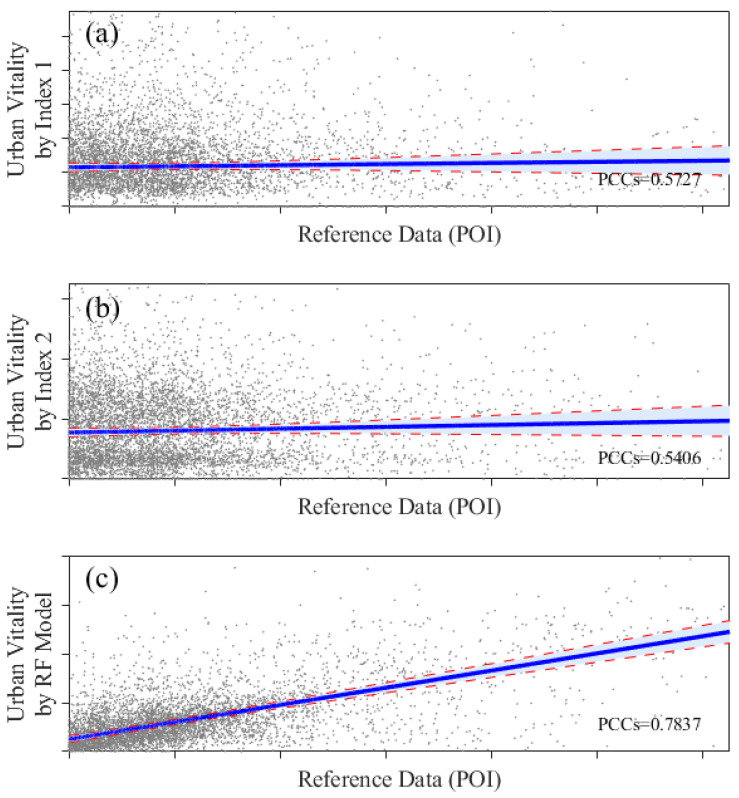
Scatter plots (gray points), regression curves (blue solid lines), and confidence intervals (red dash lines and light blue fills) of urban vitality reference data and evaluation results (normalized to 0–1), including (**a**) Index 1, (**b**) Index 2, and (**c**) Random Forest model.

**Figure 8 ijerph-20-03821-f008:**
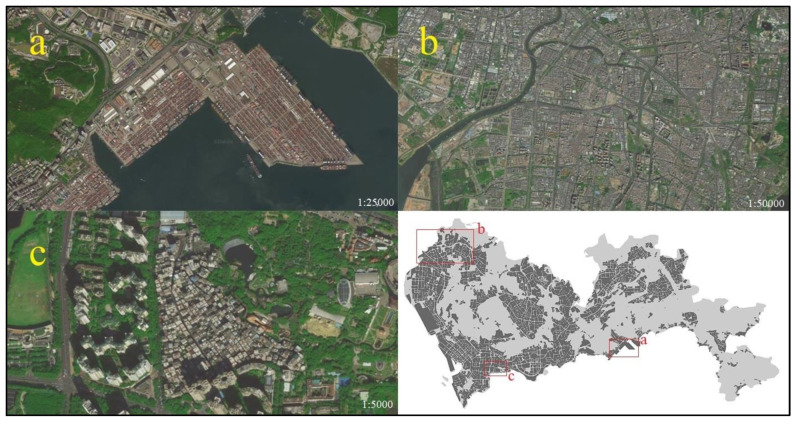
Satellite image of Yantian Port (**a**), North Baoan (**b**) and Baishizhou Street (**c**) and the locations on the Baidu Map.

**Figure 9 ijerph-20-03821-f009:**
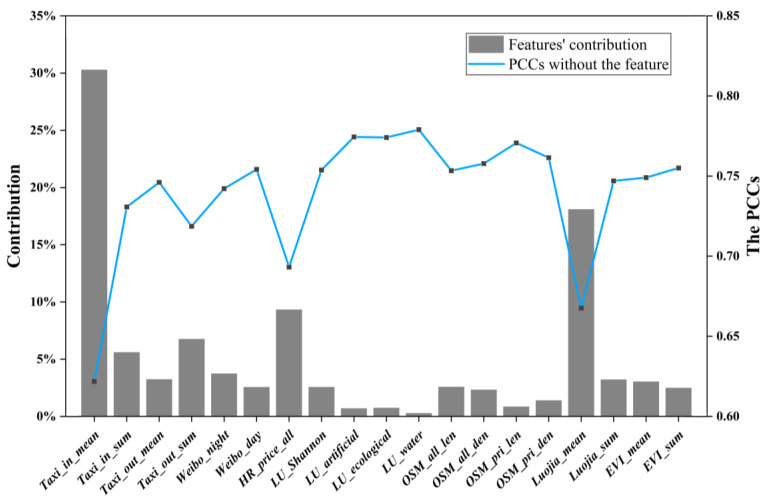
Features’ contribution in Random Forest model and the PCCs of RF models without the corresponding features.

**Figure 10 ijerph-20-03821-f010:**
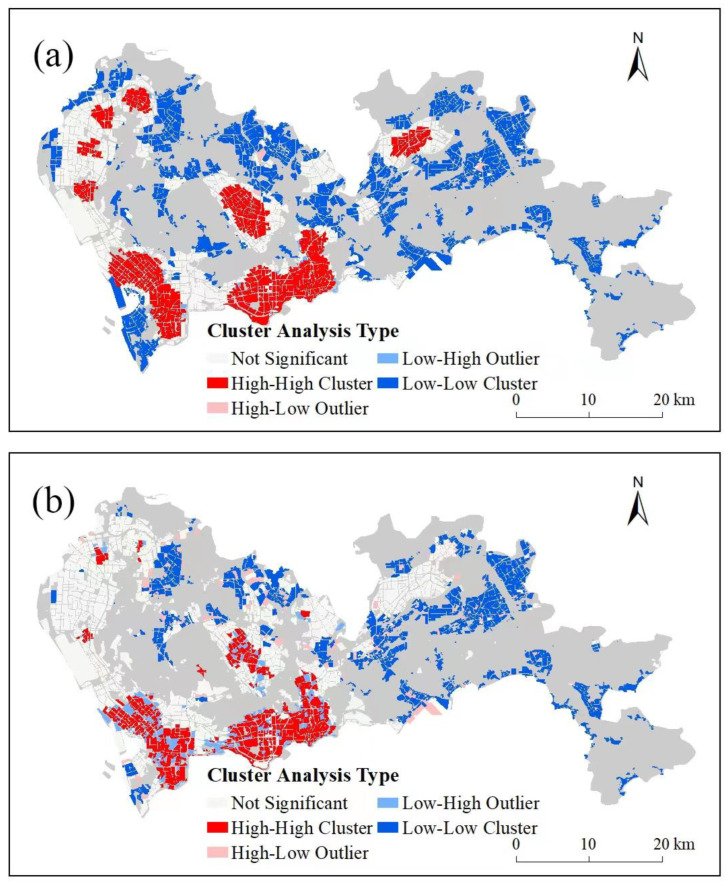
Cluster analysis results of urban vitality by POI reference data (**a**) and the Random Forest model (**b**).

**Table 1 ijerph-20-03821-t001:** Data and Features for urban vitality evaluation.

Data Source & Website	Feature	Description
Taxi trajectory https://people.cs.rutgers.edu/~dz220/data.html accessed on 30 November 2019	*Taxi_in_mean*	Average length of tracks whose end point locates in
	*Taxi_out_mean*	Average length of tracks whose start point locates in
	*Taxi_in_sum*	Total length of tracks whose end point locates in
	*Taxi_out_sum*	Total length of tracks whose start point locates in
	*Taxi_total_mean*	Average value of *Taxi_in_mean* and *Taxi_out_mean*
Weibo check-in http://m.weibo.cn accessed on 31 August 2019	*Weibo_sum*	Sum value of *Weibo_day* and *Weibo_night*
	*Weibo_day*	Number of Weibo Signing in during 7 a.m. to 6 p.m.
	*Weibo_night*	Number of Weibo Signing in during 6 p.m. to 7 a.m.
Housing rental https://shenzhen.anjuke.com accessed on 31 October 2019	*HR_price_all*	Average of all rents
	*HR_pirce_com*	Average of commercial rents
	*HR_price_res*	Average of residential rents
Land use www.globallandcover.com accessed on 31 December 2020	*LU_Shannon*	Shannon diversity index
	*LU_artificial*	Proportion of artificial lands pixels
	*LU_ecological*	Proportion of ecological lands pixels including forest, grass and cultivated lands
	*LU_water*	Proportion of waterbody pixels
OSM road https://www.openstreetmap.org accessed on 30 February 2022	*OSM_all_len*	Sum length of all roads
	*OSM_all_den*	Road density of all roads
	*OSM_pri_len*	Sum length of primary roads
	*OSM_pri_den*	Road density of primary roads
	*OSM_sec_len*	Sum length of secondary roads
	*OSM_sec_den*	Road density of secondary roads
Nighttime light http://59.175.109.173:8888 accessed on 30 November 2018	*Luojia_mean*	Average value of all Luojia pixels
	*Luojia_sum*	Sum value of all Luojia pixels
MODIS NDVI/EVI https://ladsweb.nascom.nasa.gov/search accessed on 31 December 2018	*EVI_mean*	Average value of all EVI pixels
	*EVI_sum*	Sum value of all EVI pixels
	*NDVI_mean*	Average value of all NDVI pixels
	*NDVI_sum*	Average value of all NDVI pixels

**Table 2 ijerph-20-03821-t002:** Parameters’ settings of Random Forest model.

Parameter	Meaning of Parameters	Range	Footstep
N_estimators	The maximum number of the weak learners obtained from the original dataset	10–201	5
Max_features	The number of features in each decision tree	1–20	1
Max_depth	The maximum depth of a decision tree	1–20	1
Min_samples_split	Minimum number of samples for leaf nodes	1–50	3
Min_samples_leaf	Minimum number of samples for node splitting	2–50	2

**Table 3 ijerph-20-03821-t003:** Global Moran Index of urban vitality.

	Moran’s I	Z Score
Urban Vitality Index 1	0.2068	6.70
Urban Vitality Index 2	0.1959	5.94
Random Forest Model	0.4795	15.52

## Data Availability

There were no new data created in this study. All data in this study are open-accessed online and the links can be found in the text.

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
