# Peer review of "Evaluating Urban Vitality of Street Blocks Based on Multi-Source Geographic Big Data: A Case Study of Shenzhen"

_ijerph, 2023, doi:10.3390/ijerph20053821_

Round 1
Reviewer 1 Report
This study measures urban vitality at the block level in Shenzhen, China, using indexing and ML methods, and compares the results of both methods and discusses which method better explains urban vitality. In the study, POI data is used as reference data for urban vitality and it is tested to what extent the variables used in both models explain the reference data. However, in order for this study to be published, I think the following issues should be addressed:
MAJOR ISSUES
1. The first of the main problems with the manuscript is the ability of the POI data to represent urban vitality. In the nearly a century old urban vitality literature from Jacobs to the present, many criteria are evaluated together, especially density (density of people, buildings and activities), diversity (diversity of people, activities and functions), interaction, accessibility and permeability. Therefore, the assumption that POI data alone represents urban vitality should be reconsidered. As a matter of fact, the authors express this fact in the introduction as follows (Lines 63-66):
“First, data commonly used in index method only reflect limited aspects of urban vitality, for example, POI data emphasizes on distributions of facilities and zones of economic activities (Yue et al., 2021); social media check-in data stresses people's activities (Zhu et al., 2020); taxi GPS data reflects the flows of people (Zhang et al., 2022b).”
The authors did not provide a reasonable justification for comparing the index and ML methods with reference to POI data, accepting POI data as reference data.
2. The authors explain the data they used in the study and the variables they produced from this data in Table 1. Among these variables, two sets of variables (land use and MODIS) produced using remote sensing data are based on global land cover, not urban land use. However, since the analysis unit of the authors is urban blocks, land covers such as forest and agricultural areas are outside the study area. These two variables do not give the details of urban land use, but combine them all in the artificial land legend. However, in order to measure urban vitality, it is necessary to look at the diversity of urban uses (residential, commercial, recreational, public facilities etc). Global land cover doesn't say much for urban vibrancy. In fact, the authors themselves express this fact when presenting the findings of the study (Lines 455-459).
3. When comparing results, the authors argue that the ML model is more objective, while the index method is subjective. However, in both models, the authors add the variables themselves manually and based on the PCC matrix. In doing so, another problem arises. Namely: While the authors do not include OSM features, MODIS features, and land use features in the first model because of the low PCC values with the reference data, they do not see any harm in including them in the second model. On the other hand, they included the taxi_in_mean and taxi_out_mean variables, which have very high correlation values, into the second model by saying "considering that in-features and out-features might have different influence". Therefore, the selection process of the variables in both models was done manually. For example, "taxi_total_mean" and "Weibo_sum" variables added manually to the first model are considered redundant for the second model. This raises the question of the reliability of the results.
4. Looking at the correlation coefficients in Figure 5, it is understood that there is a very high positive correlation between the taxi_in_mean, taxi_out_mean and taxi_total_mean variables. In the same figure, low positive correlation was measured between the taxi_in_mean / taxi_out_mean variables and the reference data, while a negative correlation was observed between the taxi_total_mean variable and the reference data. The measurement results should be checked. (If there is a positive correlation between A and B and a positive correlation between A and C, a positive correlation is expected between B and C).
5. What is the rationale for the formulations in Table 2? How and on what basis were these formulas produced? Why isn't a weighting preferred in indexing? How did the results of these indexes compare with Figure 3? (L.307-308) There is no satisfactory answer to these questions in the article.
MINOR ISSUES
1. Instead of sharing links in the text (L.155, 164, 170,175,185,190,202), web links can be given in a seperate table or integrated into Table 1.
2. In my opinion, there is no need for the phrase “quantitatively” in the title. The analysis of big data is expected to be quantitative anyway.
3. Since the blocks are the basic analysis unit (Lines 138-145), the authors should give information about the minimum, maximum and average block sizes in the study area.
4. What is cell size of each pixel for raster data that the authors used? (L.221-224)
5. The word of “research” should not be used in plural form (Lines 41, 289, 454). (The word "studies" may be preferred instead).
6. Authors should explain whether they used the total number of POIs in each block or the density of POIs (The number of POIs / Block Size) as variables in the analysis. (L.212)
7. References 30 and 32 refer to the same reference. One of them (number 30) must be removed from the list. The correct use in the text should be (Pakoz and Işık, 2022). (Please correct Line 43).
8. In the discussion / conclusion section, it is expected that this study will be compared with previous studies on urban vitality in Shenzhen in terms of methods, approaches and findings.
Reviewer 2 Report
Thank you for giving me this opportunity to read the manuscript entitled "Quantitatively Evaluating Urban Vitality of Street Blocks Based on Multi-source Geographic Big Data: A Case Study of Shenzhen". The topic of this manuscript is interesting and would be a good contribution to this field. I think it could be considered for publication in IJERPH once the following issues are addressed.
1. Please replace the keywords that already appear in the manuscript's title with close synonyms or other keywords, which will also facilitate your paper being searched by potential readers.
2. "Legend", “Scale”, and "Compass" should be added to the map of China in Figure 1.
3. Line 55, “, social media check-in data (He et al., 2018, Zhu et al., 2020) …”: a paper titled “Dynamic assessment of PM2. 5 exposure and health risk using remote sensing and geo-spatial big data” is suggested to be added as a reference to support the statement here.
4. “Local-based service data” should be added as another geolocation data, and the following two references could be added as references: (1) Observed inequality in urban greenspace exposure in China. (2) How does urban expansion impact people’s exposure to green environments? A comparative study of 290 Chinese cities.
5. “Legend” should be added to Figure 2.
6. Lines 185-186: More detailed information regarding the Python web crawler should be provided.
7. Section 2.4. the spatial matrix used for Local Moran index should be provided.
8. Limitation section should be added as a sub-section to the Discussion.
9. Some grammatical errors exist in the manuscript. Therefore, a critical review of the manuscript's language will improve its readability.
Round 2
Reviewer 1 Report
Thank you for the response and efforts to improve the manuscript. However, I do still have a major concern about the revised version of the manuscript.
In the first version of the manuscript, authors say (L. 262-264) that “Taxi_total_mean’s PCCs was negative, while the other three features’ PCCs were positive. Therefore, when building indexes, feature Taxi_total_mean served as the negative term while other features served as the positive terms.”
After they revised the manuscript, they corrected the correlation matrix and it is understood that Taxi_total_mean’s PCCs was positive. But they did not change the formulas they used in Table 2. I am not convinced by the following explanation in the response letter:
“We found that when Taxi_total_mean was the subtraction or division, and other features were the addition or multiplication, the indexes performed better, but it is hard to explain this.”
Because there is no rationale for subtraction or division of Taxi_total_mean if they base on the PCCs.
My recommendation is to simplfy the index construction process and make each index alternative explainable. And they also need to revise the sections of Results and Discussion accordingly.
(There are some grammar mistakes in the revised manuscript also. Please check them again).
Reviewer 2 Report
Thank you for giving me this opportunity to read the revised version of the manuscript titled "Evaluating Urban Vitality of Street Blocks Based on Multi-source Geographic Big Data: A Case Study of Shenzhen", and for the detailed responses to my earlier comments. I am satisfied with this revised version, and I think it is acceptable now.
Author Response
Sincerely thank you for your feedback on our resubmitted manuscript (ijerph-2192539). We are very happy that you are satisfied with our revisions and responses. We appreciate for all your review works and your helpful suggestions. With your comments, our manuscript has been thoroughly improved, and your contribution to our manuscript cannot be ignored. It's a pleasure to cooperate with you.